# Diagnosis of Rotor Component Shedding in Rotating Machinery: A Data-Driven Approach

**DOI:** 10.3390/s24134123

**Published:** 2024-06-25

**Authors:** Sikai Zhang, Qizhe Lin, Jiayao Lin

**Affiliations:** 1Institute of Big Data and Information Technology, Wenzhou University, Wenzhou 325027, China; 13566161223@139.com (S.Z.); linqz@symacnc.com (J.L.); 2College of Mechanical and Electrical Engineering, Wenzhou University, Wenzhou 325027, China

**Keywords:** fault diagnosis, rotating machinery, feature extraction, principal component analysis (PCA), linear discriminant analysis (LDA)

## Abstract

The potential for rotor component shedding in rotating machinery poses significant risks, necessitating the development of an early and precise fault diagnosis technique to prevent catastrophic failures and reduce maintenance costs. This study introduces a data-driven approach to detect rotor component shedding at its inception, thereby enhancing operational safety and minimizing downtime. Utilizing frequency analysis, this research identifies harmonic amplitudes within rotor vibration data as key indicators of impending faults. The methodology employs principal component analysis (PCA) to orthogonalize and reduce the dimensionality of vibration data from rotor sensors, followed by k-fold cross-validation to select a subset of significant features, ensuring the detection algorithm’s robustness and generalizability. These features are then integrated into a linear discriminant analysis (LDA) model, which serves as the diagnostic engine to predict the probability of rotor component shedding. The efficacy of the approach is demonstrated through its application to 16 industrial compressors and turbines, proving its value in providing timely fault warnings and enhancing operational reliability.

## 1. Introduction

Rotating machinery is widely used in various industrial applications, and the operation of rotating machinery, such as compressors, gas turbines, and steam turbines, is characterized by high rotational velocities and significant momentum. This can lead to the loosening or detachment of rotor components, posing a substantial threat to operational safety and often resulting in subsequent damage to stationary and non-stationary blades or impellers. Therefore, in all kinds of faults, rotor parts falling off is a major safety hazard. And accurate prediction of rotor component detachment, or shedding faults, is essential for implementing condition-based maintenance strategies [1], which are paramount for both safety and economic efficiency.

Traditional methods for rotor fault detection are categorized into time-domain and frequency-domain analyses. Time-domain analysis commonly employs orbit analysis for fault detection [2], utilizing data from two radial displacement sensors to monitor changes in the rotor’s orbit. Features extracted from the orbit plot, such as the number of inner loops and changes in orientation, are recognized as sensitive indicators of rotor crack faults [3]. The work of Chang et al. [4] enhanced the diagnosis of various rotor failure types by pre-processing orbit data with a bivariate histogram and applying fractal theory to extract fractal dimension and lacunarity. In the frequency domain, the amplitudes of harmonic components are typical features [5], with specific harmonic amplitudes indicating rotor faults. For instance, an increased amplitude of the second-order harmonic of rotational speed is a well-established indicator of rotor misalignment [6]. Critical harmonics may be identified through observation by comparing spectra of normal and faulty rotors [7] or derived mathematically from dynamic models, necessitating prior knowledge of rotor dynamics [8]. Yu and Zhou [9] utilized a one-dimensional residual convolutional autoencoder (1D-RCAE) for gearbox fault diagnosis, extracting key features from vibration signals. Kalista et al. [10] proposed a method using vibration sensors and a notch filter to generate precise rotor orbit shapes, aiding rotor system analysis. Zhao et al. [11] introduced a demodulation technique for vibration monitoring of rolling bearings under variable speeds, enhancing signal analysis in machinery health monitoring.

Studies concerning principal component analysis (PCA) demonstrated its utility across various domains. Mao Ge et al. [12] proposed a technique for bearing fault diagnosis using local robust PCA and multi-scale permutation entropy, significantly enhancing fault diagnosis accuracy and reliability. Similarly, Shuna Jiang et al. [13] employed a learning vector quantization neural network and kernel PCA for fault diagnosis within proton exchange membrane fuel cell (PEMFC) water management subsystems, highlighting the benefits of integrating neural networks with PCA.

Zahoor Ahmad et al. [14] applied informative ratio PCA to fault diagnosis in multistage centrifugal pumps, achieving notable advancements in detection capabilities. Keshun You et al. [15] utilized a hybrid neural network with PCA for bearing fault diagnosis, showcasing the advantages of combining these methods. Chenpeng Liu et al. [16] introduced dynamic PCA and genetic algorithm for feature selection in industrial process fault diagnosis, furthering the field’s methodologies.

Bin Chen et al. [17] explored the integration of generative adversarial network (GAN) technology with PCA for bearing fault diagnosis, illustrating the potential of GANs in this context. These studies collectively emphasize the versatility of PCA and its enhancement when paired with other techniques. Overall, PCA offers advantages such as dimensionality reduction, feature extraction, noise reduction, visualization, compatibility with other techniques, and interpretability, making it a powerful tool for various data analysis and fault diagnosis tasks in engineering and scientific domains.

Advancements in machine learning for rotor fault diagnosis have been significant. Feng et al. [18] introduced the deep reconstruction transfer convolutional neural network (DRTCNN) for rolling bearing fault diagnosis, merging deep learning with transfer learning and CNNs. Rauber et al. [19] assessed machine learning methods for fault diagnosis using vibration signals. Tong et al. [20] developed an intelligent fault diagnosis method for rolling bearings using GADF and IDARN, while Azamfar et al. [21] integrated multisensor data fusion with 2-D CNN and motor current signature analysis for gearbox fault diagnosis. Maurya et al. [22] focused on condition monitoring by combining EMD-based local energy features with a DNN, leading to effective machine condition classification. He and He [23] proposed a hybrid deep signal processing approach for bearing fault diagnosis, enhancing the accuracy of diagnosis by integrating deep learning algorithms with traditional signal processing methods. Additionally, Jeon et al. [24] contributed to the field by determining the optimal vibration image size for fluid-film rotor-bearing system diagnosis using convolutional neural networks (CNNs), thereby facilitating more effective image-based fault diagnosis.

Another notable advancement is feature fusion and selection, where machine learning algorithms like support vector machines (SVMs) and genetic algorithms (GAs) are employed to fuse relevant features and optimize feature selection [25,26,27,28,29,30,31]. This approach has demonstrated efficacy in improving fault diagnosis accuracy by focusing on the most informative aspects of the data.

Another significant advancement in the field is the utilization of feature fusion and selection techniques. Researchers have delved into the application of machine learning algorithms, including support vector machines (SVMs) and genetic algorithms (GAs), to effectively integrate relevant features and streamline the feature selection process [25,26,27,28,29,30,31,32]. These methodologies have demonstrated efficacy in bolstering the precision of fault diagnosis by concentrating on the most informative data dimensions.

Ensemble learning techniques have also played a pivotal role in fault detection and classification performance. These techniques combine multiple models, such as random forests and classifier ensembles, to bolster the robustness and reliability of fault diagnosis systems for rotating machinery [32,33,34,35].

Concurrently, Jian Cen et al. [36] provided an exhaustive review of data-driven approaches for machinery fault diagnosis using machine learning algorithms. The review suggested exploring transformer neural networks, which are based on pure attention mechanisms, as a promising direction for future research, particularly due to their parallel computation advantages over long short-term memory (LSTM) networks. The review also identified areas for improvement in transferability, federated transfer learning, and addressing strong noise backgrounds in machinery fault diagnosis methods reliant on machine learning algorithms, offering valuable insights for future research directions. This includes the use of supervised learning algorithms (e.g., support vector machines, decision trees, random forests) for classification and regression tasks using labelled training data, unsupervised learning algorithms (e.g., clustering, principal component analysis) for anomaly detection and pattern recognition without labelled data, and deep learning algorithms (e.g., convolutional neural networks, recurrent neural networks) for feature learning and complex pattern recognition within large datasets.

Despite considerable research on rotor faults like rub, crack, and misalignment, the issue of shedding fault has garnered less attention. Moreover, feature selection in traditional time- and frequency-domain analyses often requires human observation or prior knowledge, which may not be feasible in complex rotor operating environments where rotor parameters are uncertain. This study aimed to address the gap left by traditional methods by proposing a data-driven fault diagnosis method that integrates principal component analysis (PCA) and linear discriminant analysis (LDA) to effectively identify rotor component shedding. The objective of this research was to demonstrate the superiority of the PCA-LDA approach over existing methods, such as independent component analysis (ICA) and support vector machines (SVM), in terms of classification accuracy and robustness to data variability.

The research was structured into five distinct phases, as depicted in Figure 1. In the pre-processing stage, traditional orbit and harmonics analyses were applied to extract damage-sensitive features. To enhance fault detection performance, PCA was performed on the vibration data. In the normalization stage, the appropriate normalization method for this scenario was discussed and analysed. A portion of the data was utilized for determining the optimal number of features and for feature selection. A modified cross-validation approach was introduced and implemented during these two stages. In the classification stage, the data used for feature selection served as the training data for the linear discriminant analysis (LDA) model, while the remaining data were employed for testing. The proposed method was applied to 16 operating compressors and turbines in China. The results highlight the method’s effectiveness in distinguishing between normal and faulty rotors and in providing timely early warnings for rotor component shedding faults, underscoring the potential of data-driven methods in predictive maintenance. This study also emphasizes the potential utility of radial eddy current displacement sensor data in rotor parts’ shedding fault detection, setting a benchmark for future research endeavours in this domain.

## 2. Methodology and Model Selection

Dimensionality reduction is a fundamental strategy in the realm of data analysis, and principal component analysis (PCA) is a preeminent technique within this field. A robust dimensionality reduction method must efficiently compress data while maximizing the retention of data variances. The essence of PCA lies in its methodical identification of a linear basis that captures the essence of the reduced dimensionality space. In this section, we elucidate the linear discriminant analysis (LDA) methodology and detail its implementation as a diagnostic tool for binary classification challenges. Additionally, we provide a rationale for not incorporating fault detection and analysis (FDA) techniques.

### 2.1. Principal Component Analysis (PCA)

PCA is a revered statistical technique, highly esteemed for its ability to succinctly reduce the dimensionality of extensive datasets, all the while striving to maintain their intrinsic variability. This is achieved by projecting the original data onto a new set of uncorrelated variables, termed principal components. These components are strategically ordered such that the foremost components encapsulate the most significant variations present in the original dataset.


**Standardizing the Data:**


The preliminary step in PCA involves standardizing the dataset, a critical phase that aligns each feature to a zero mean and unit standard deviation. This normalization is essential due to PCA’s acute sensitivity to the variances of the initial variables.

Mathematically, for a dataset *X*, comprising *n* observations and *p* variables, the standardized variable *Z_ij_* is derived from *X_ij_* as follows:(1)$$Z_{ij}=\frac{X_{ij}-\mu_{i}}{\sigma_{i}}$$
where μ and σ are the mean and standard deviation of the variable *X*, respectively.


**Computing the Covariance Matrix:**


With the standardized data, we proceed to calculate the covariance matrix, an essential tool that delineates the variance and covariance among variables. The covariance matrix Ʃ for the standardized data matrix *Z* is articulated as:(2)$$\sum =\frac{1}{n-1}Z^{T}Z$$

This matrix is pivotal in discerning the inter-variable variance dynamics.


**Obtaining Eigenvalues and Eigenvectors:**


Subsequently, we compute the eigenvalues and eigenvectors of the covariance matrix. The eigenvectors, corresponding to the principal components, identify the directions of maximum variance, while their associated eigenvalues measure the magnitude of variance in these directions.

Given Ʃ as the covariance matrix, the eigenvalue equation is formulated as:(3)∑ν=λν
where *ν* represents the eigenvector and *λ* represents the eigenvalue.


**Selecting Principal Components:**


The selection of principal components hinges on the eigenvalues’ magnitudes, with precedence given to those with the largest eigenvalues, as they embody the most influential underlying data patterns.

The retention of principal components is often dictated by the cumulative explained variance ratio, ensuring that a threshold percentage (e.g., 95%) of the total variance is conserved.


**Transforming the Data:**


The original data are subsequently translated into a novel feature space, demarcated by the chosen principal components. This transformation, while reducing feature count, dutifully preserves the original data’s quintessential attributes.

The transformed dataset *Y* is rendered as:(4)Y=Zν
where ν denotes the matrix comprising selected eigenvectors.

### 2.2. Linear Discriminant Analysis (LDA)

Following the PCA transformation, linear discriminant analysis (LDA) is applied to the streamlined feature set for fault classification. LDA operates by maximizing the ratio of between-class variance to within-class variance in the PCA-transformed data, thereby enhancing class separability. The synergistic application of PCA for feature extraction and LDA for classification harnesses the strengths of both methods, leading to more precise and efficient fault diagnosis.

LDA is a supervised learning method designated for classification, aiming to identify a linear combination of features that optimally distinguishes between classes of objects or events.

**Assumption of Normality:** LDA posits that the data from each class are normally distributed, simplifying computations and ensuring optimal class separation in the transformed space.

**Maximizing Class Separability:** LDA works by finding the linear combinations of features (known as discriminant functions) that maximize the ratio of between-class variance to within-class variance. This ensures that the classes are as distinct as possible in the new feature space.

**Computing the Discriminant Functions:** The discriminant functions are computed using the eigenvectors of the scatter matrices. The between-class scatter matrix (*S_B_*) and the within-class scatter matrix (*S_W_*) are defined as:(5)$$S_{B}=\sum_{i=1}^{k}N_{i}(\mu_{i}-\mu ){\left(\mu_{i}-\mu \right)}^{T}$$
(6)$$S_{W}=\sum_{i=1}^{k}\sum_{j=1}^{N_{i}}(X_{ij}-\mu_{i}){\left(X_{ij}-\mu_{i}\right)}^{T}$$
where μ is the mean vector of the total samples, $\mu_{i}$ is the mean vector of the samples in class i, and *N_i_* is the number of samples in class i.

**Projection onto Discriminant Axes:** The data are projected onto the new axes defined by the discriminant functions. This projection enhances the separation between classes, making it easier to classify new data points.

By integrating LDA with PCA, we leverage the ability of PCA to reduce dimensionality and noise and the power of LDA to enhance class separability. This combination improves the robustness and accuracy of the fault diagnosis system, making it more effective in identifying and predicting rotor component shedding.

In summary, the PCA-based feature extraction method is a crucial step in our data-driven approach, allowing us to effectively process and analyse the complex vibration data from rotating machinery. This method not only simplifies the data but also enhances our ability to detect and diagnose rotor component shedding at an early stage. The integration of LDA further refines the classification process, ensuring that the system can accurately differentiate between normal and faulty conditions, leading to timely and reliable fault detection and diagnosis.

### 2.3. Comparison with Existing Data-Driven Methods

In the rapidly evolving landscape of data-driven fault diagnosis techniques, a multitude of methods has emerged. Herein, we provide a comparative analysis of our selected methodology against two notable data-driven approaches: independent component analysis (ICA) and support vector machine (SVM).

**Independent Component Analysis (ICA):** ICA is a computational technique used to separate a multivariate signal into independent, non-Gaussian components. While ICA has been successful in various applications, it has some limitations when applied to our rotor system dataset. ICA requires the assumption of statistical independence and non-Gaussianity of the source components, which may not always be applicable or easily verifiable in real-world datasets. Additionally, ICA does not inherently provide a classification mechanism, which is essential for our diagnostic needs.

**Support Vector Machine (SVM):** SVM is a powerful supervised learning model known for its effectiveness in high-dimensional spaces and its ability to handle non-linear relationships through kernel tricks. However, SVM’s performance is highly dependent on the selection of kernel functions and regularization parameters, which can necessitate extensive hyperparameter tuning. Furthermore, SVM may not scale as efficiently with large datasets as LDA, which is a critical consideration given the extensive data generated by our rotor system sensors.

**Rationale for Choosing PCA and LDA:** After meticulous consideration and evaluation, PCA and LDA were chosen over ICA and SVM for several compelling reasons. PCA offers an efficient mechanism for dimensionality reduction while preserving data’s essential structure, crucial for handling the substantial volumes of data collected from our sensors. The linear nature of PCA also facilitates the straightforward interpretation of results, aiding in the diagnostic process.

LDA was selected for its effectiveness in classification tasks, particularly when dealing with binary classification problems such as distinguishing between normal and faulty rotor conditions. LDA’s ability to find a linear combination of features that maximizes class separability makes it well suited for our application. Additionally, LDA’s computational efficiency and scalability make it a practical choice for analysing large datasets.

In conclusion, while ICA and SVM are powerful tools in their own right, PCA and LDA were chosen for their combined strengths in dimensionality reduction and classification, which are particularly well suited to the requirements of our rotor system fault diagnosis task.

## 3. Experimental Setup and Data Processing

### 3.1. Sensors’ Installation and Data Description

The vibration data were captured using four eddy current displacement sensors, as depicted in Figure 2. Pairs of sensors were strategically installed at the coupling and non-coupling ends of the rotor to measure radial displacements in orthogonal directions labelled X1, Y1, X2, and Y2. Comprehensive datasets were compiled from 16 rotors, designated M1 to M16, sourced from compressors and steam turbines operating in industrial settings in China.

The eddy current displacement sensor used in the experiment is a universal eddy NCDT3005-type current displacement sensor produced by MICRO-EPSILON GMBH in Germany. This sensor has the advantages of a simple structure, wide frequency response range, high sensitivity, and strong anti-interference ability and is widely used in the measurement of vibration displacement of rotating machinery rotor. The basic parameters are shown in Table 1.

The sensor is powered by a 15 V DC power supply, and the displacement signal measured by the sensor probe is adjusted by the amplifier to output 0~10 V DC voltage. The 0~10 V dynamic DC voltage signal output by the sensor is recorded by the data acquisition system. The high sampling rate data acquisition system NI PCI-6221 produced by NI Company (Austin, TX, USA) was used in the experiment. It includes an NI PCI-6221 data acquisition card, NI CB-68LP terminal, and NI SHC68-68-S dedicated low-noise data cable. The main performance parameters of the data acquisition card are shown in Table 2.

An equal number of faulty and normal rotors, totalling eight of each, were included in the study. The data structure is illustrated in Figure 3, with each rotor providing five temporal segments of data, sequentially denoted as a, b, c, d, and e. The latest and earliest measurements are indicated by ‘a’ and ‘e’, respectively. For faulty rotors, component shedding occurred within half a year post the ‘a’ measurement, whereas normal rotors remained failure-free for one year beyond the ‘a’ measurement. During each time segment, the sensors concurrently recorded approximately 200–300 waves. Each wave was comprised of 1024 data points, corresponding to the rotor’s radial displacement across 32 revolutions. An exemplar wave is portrayed in Figure 4.

### 3.2. Pre-Processing and Feature Extraction

As it cannot be guaranteed that the installation positions of the sensors on all the rotors are the same, the data from different rotors do not have comparability. In addition, as the orthogonality of the directions X1 and Y1 (or X2 and Y2) cannot be guaranteed also, the correlation between the two directions increases the difficulty of the analysis. Therefore, we provide an in-depth examination of the application of principal component analysis (PCA) for feature extraction from the vibration data. To solve the two problems, the principal component analysis (PCA) was applied on the waves of the coupling and non-coupling end, respectively. The two waves measured at the same time on X1 and Y1 (or X2 and Y2) were formed as a pair of waves of the coupling end (or the non-coupling end). Through PCA, the pair of waves were transformed into two principal components, PC1 and PC2. PC1 was always in the maximum variance direction, while PC2 was always orthogonal to PC1. An example is given in Figure 5. The horizontal and vertical values of the orbit in Figure 5a are given by the pair of waves in the X1 and Y1 directions, respectively. As a wave measures 32 revolutions, there were 32 rings in the orbit. After PCA, the orbit was rotated (Figure 5b), so that the values in the horizontal direction had the maximum variance and the values in the vertical direction were orthogonal to the values in the horizontal direction.

PCA was applied to both the coupling and non-coupling ends, so two pairs of PC1 and PC2 were obtained. The harmonic amplitudes of the four principal components were chosen as the damage-sensitive features. As the wave had 1024 data points, which were sampled from 32 revolutions, the sampling frequency was given by
(7)$$F_{S}=\frac{1024\times f}{32}$$
(8)$$F_{r}=\frac{f}{32}$$
where *f* Hz is the rotation speed. The spectra of PC1 and PC2 were obtained by the fast Fourier transform (FFT), and the frequency resolution of the spectra was given by

It should be noticed that the sampling frequency in this case was varying with the rotation speed. The sampling technique is called angular sampling, which samples the data at a constant angular interval rather than a constant time interval. As the first point of the spectrum was at 0 Hz, the 33rd point of the spectrum was exactly at f Hz according to (8). The nth harmonic, which was at *n* times the rotation speed, was at the 1 + 32nd point of the spectrum. According to the Nyquist–Shannon sampling theorem, we only analysed the spectrum from 0 Hz to Nyquist frequency, which was from the 1st point to the 512th point. Therefore, the maximum of *n* was 15 and there were 15 harmonic amplitudes that could be extracted from the spectrum. The peaks of the normal spectrum were regularly distributed at the harmonic frequencies, which is shown in Figure 6a. In some special conditions, e.g., the rotating speed was less than 10 rpm, the peaks were generally not at the harmonic frequencies, which is shown in Figure 6b. These types of spectra were deemed as abnormal, so were removed from the raw data.

In addition to the 15 harmonic amplitudes, the root mean square (RMS) of the wave was also used as the damage-sensitive features. The RMS of the wave indicates the equivalent steady-state energy, and the RMS is computed by
(9)$$x_{RMS}=\sqrt{\frac{1}{N}\sum_{i=1}^{N}{\left|\nu_{i}\right|}^{2}}$$
where n is the number of data and $\nu_{i}$ is the data in the wave.

### 3.3. Data Normalization

Normalization of the data is a critical step following the extraction of *RMS* and harmonic amplitude features. It is essential to standardize the feature set to improve the classification model’s performance. Normalization mitigates the impact of statistical properties such as mean and standard deviation, which could otherwise distort the model training process. By eliminating these variances, we reduce the potential for misleading the classification model’s training.

The fluctuation range, defined as the difference between the maximum and minimum values of each feature, was scrutinized for its correlation with rotor failure. It was expected that the fluctuation range of the faulty rotors during the five time segments may be different from the normal rotors. The fluctuation range vector of one feature can be defined as
(10)$$W_{i}={\left[w_{1},\ldots ,w_{16}\right]}^{T}$$
where $w_{i}$ is the fluctuation range of the feature for $m_{i}$ rotor. The correlation between the fluctuation range vector and the rotor failure can be evaluated by the biserial correlation coefficient (BCC) given by
(11)$$r=\frac{m_{1}-m_{0}}{S_{n}}\sqrt{\frac{n_{1}n_{0}}{n^{2}}}$$
where $m_{1}$ is the mean value on the W for all $w_{i}$ from the faulty rotors, $m_{0}$ is the mean value on the W for all $w_{i}$ from the normal rotors, $n_{1}$ is the number of the faulty rotors, $n_{0}$ is the number of the normal rotors, *n* is the number of all rotors, and $S_{n}$ is the standard deviation of the W. We computed the BCC for each feature; the average BCC was 0.5111, which implied the fluctuation range had a moderate correlation with the rotor failure. Thus, we determined the fluctuation ranges should not be removed from the features.

Although the fluctuation ranges were related to the rotor failure, the relationship between the minimum values and the rotor failure was weak. An example is illustrated in Table 3. It was found that the minimum value varied largely within the failure or normal rotors but had no significant difference between the failure and normal rotors, which implied the minimum values were not related to the failure. Thus, the second statistics under investigation were the minimum value of the feature. Similar to the analysis of the fluctuation range, the average BCC between the minimum value and the rotor failure was obtained; it was 0.2087. The weak correlation suggested that the minimum value could be removed. Based on the analysis of the fluctuation range and the minimum value, the normalization of the feature x was given by
(12)z=x−minx
where min x is the minimum element of the feature x. The normalization shifted all features to 0 minimum but kept the original fluctuation ranges.

### 3.4. Feature Selection

The rotor system provided four principal components, each yielding 15 harmonic amplitudes and one *RMS* feature, totalling 64 damage-sensitive features. To prevent overfitting in the machine learning models, it was necessary to select an optimal subset from these 64 features. The number of features in this subset was determined through grid search, following these steps:Randomly select *N* features from the 64 available features.The 9-fold cross-validation is performed through LDA models with the selected features.Compute the misclassification rate, which is the proportion of misclassified instances.Repeat steps 1 to 3, 1000 times, and calculate the mean misclassification rate.Vary the value of *N* and repeat the process.

The feature selection leveraged data from a diverse set of rotors, encompassing both faulty (M1, M2, M7, M9, M13) and normal (M3, M4, M6, M8) conditions. The remaining rotor data were reserved for validating the classification model, as detailed in Section 4. The data of faulty rotors in ‘a’, ‘b’, and ‘c’ time segments were categorised as faulty data, and all other data were categorised as normal data. This categorization resulted in the formation of 14 distinct datasets, as outlined in Table 4. Thus, the five faulty datasets were from the five faulty rotors, while the nine normal datasets were from the nine rotors including both the faulty and the normal rotors.

Utilizing 9-fold cross-validation, eight folds of data were allocated for model training, with the remaining fold reserved for validation. The eight training folds divided by the traditional random resampling technique were very likely to contain the waves from all nine rotors. To evaluate the generalisability of the classification model, the training and testing data in the cross-validation should be two mutually exclusive subsets. However, as the data within each rotor were highly correlated, the random resampling led to both the training and the testing data including the information of all rotors; so, the two subsets were mutually inclusive. A simplified 3-fold cross-validation example is illustrated in Figure 7a. The three blocks enclosed by thick black lines are the data of three rotors, and the data within the block are highly correlated. Although the training dataset was a subset of all data, the training dataset contained all the information of the three rotors. Therefore, the classification model trained by the training subset did not have much difference with the model trained by all the data. For this reason, the model generalisability in this case could not be evaluated by the traditional random resampling cross-validation. To solve this problem, the data in one fold were only from one rotor, which led to the training data always excluding the data from one rotor. In Figure 7b, the model was trained by the data from M1 and M3, while it was tested by the data from M2.

Figure 8 compares the performance of the three types of 9-fold cross-validation, which are:Red line: the nine folds were divided by the traditional random resampling technique, in which eight folds were used for training and one fold was used for testing.Green line: the nine folds were divided by the traditional random resampling technique, in which all nine folds were used for training and one fold was used for testing.Blue line: the nine folds were corresponding to the data of the nine rotors, in which eight folds were used for training and one fold was used for testing.

**Figure 8 sensors-24-04123-f008:**
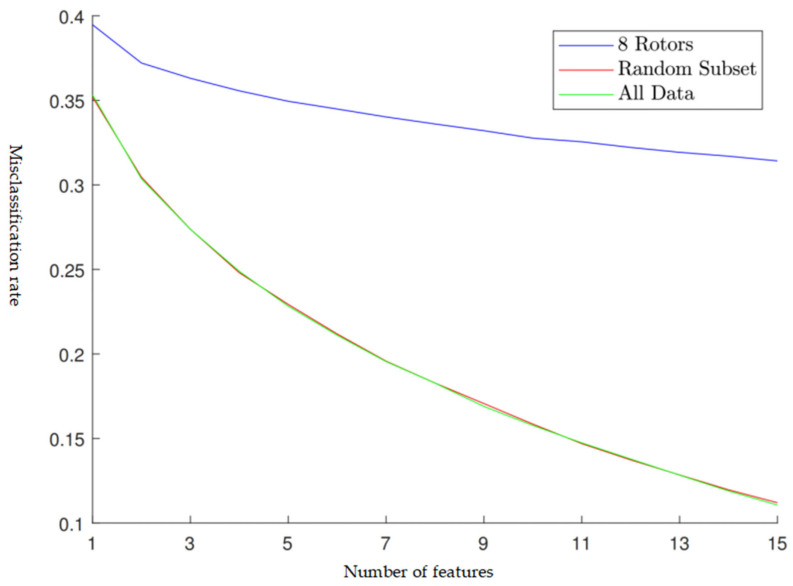
The misclassification rate for the three types of 9-fold cross-validation under the different numbers of features. The LDA models in the cross-validation were trained by the data of eight rotors, a random subset of the data of the nine rotors, and the data of all nine rotors.

It is shown in Figure 8 that the red line and the green line are very similar, which verifies the traditional cross-validation was not feasible in this case. On the blue line, the descent rate is slow, down at about 10 features, which means there was not much improvement in increasing the feature number after 10 features. Therefore, the number of the features in the optimal subset was determined as 10. After the *N* was set to 10, steps 1 to 3 in the grid search procedure were repeated 105 times to select the optimal 10 features that had the lowest misclassification rate. The optimal features selected are given in Table 5.

## 4. Classification Results and Discussion

The data of the nine rotors in the feature selection were applied for the LDA model training. The rest of the rotors, which were M10, M11, M12, M14, M15, and M16, were used for the LDA model testing. M5 was absent as the spectra of M5 in all five time segments were abnormal. The rotors used for training and testing are given in Table 6.

The selected 10 optimal features were given into the LDA model, which was trained through MATLAB function fitcdiscr. The classification results of the training and the testing are shown in Figure 9. The figures generally show five coloured circles in order. As abnormal waves were removed in the pre-process, some colours may be missing in the rotor, e.g., yellow in M1. All abnormal waves are listed in Table 7. Each colour represents a time segment, and a circle is the classification result of a wave. For example, a blue circle at ‘1’ stands for a wave in ‘a’ time segment and was classified into the faulty group by the LDA model. The values in the legend, which are the proportions of the waves in the faulty group, were used as indices to predict the health condition of the rotors. The higher the number in the colour of the legend, the more likely the rotor failed in the corresponding time segment. For the normal rotors, the indices in all five time segments were lower than 0.3. For the faulty rotors, at least two indices were higher than 0.3. Therefore, the threshold could be set as 0.3 to diagnose whether the parts’ shedding failure will happen in the rotor. In addition, for the faulty rotors, the indices of ‘a’ and ‘b’ time segments (i.e., in blue and red) were generally higher than the indices of ‘d’ and ‘e’ time segments (i.e., in magenta and green). As ‘a’ time segment was closest to the failure and ‘e’ time segment was farthest from the failure, the indices correctly indicated that the time segments were close to the failure.

## 5. Conclusions

This paper introduces a data-driven methodology specifically designed to detect rotor component shedding faults at an early stage in rotating machines. Through a comprehensive pre-processing analysis, we harnessed the power of PCA to effectively process the wave data. Our methodological approach includes a novel normalization technique, identified after a meticulous BCC analysis of fluctuation range and minimum value statistics. To enhance the robustness of our model, we proposed a modified cross-validation strategy, which outperforms the conventional random resampling cross-validation in terms of feature selection and model generalizability.

Utilizing this refined cross-validation process, we successfully identified an optimal subset of features from a pool of 64 harmonics and RMS values. The LDA model, augmented with these selected features, has proven to be highly effective in detecting faulty rotors. The initial analysis presented herein not only underscores the potential of radial eddy current displacement sensor data in the context of rotor component shedding fault detection but also establishes a benchmark for future research endeavours in this domain.

## Figures and Tables

**Figure 1 sensors-24-04123-f001:**
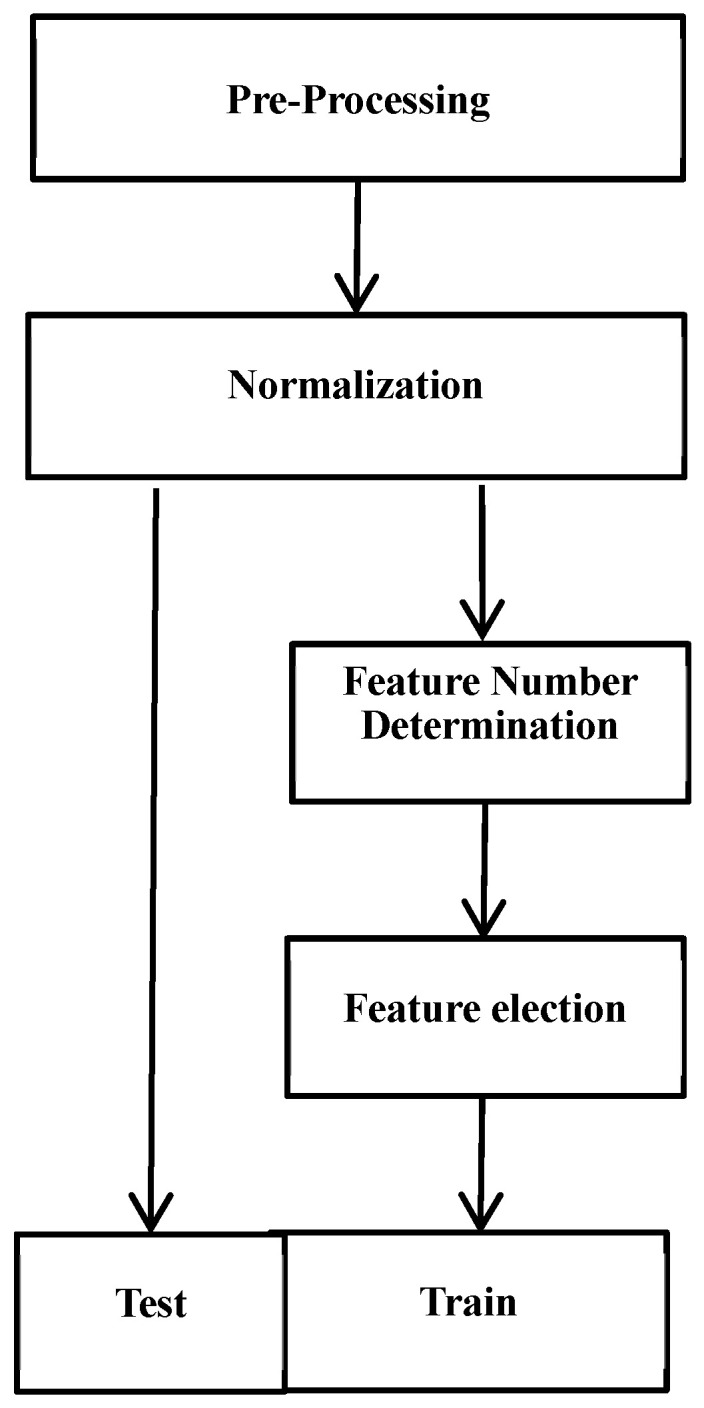
The framework of research on diagnosis of rotor component shedding fault.

**Figure 2 sensors-24-04123-f002:**
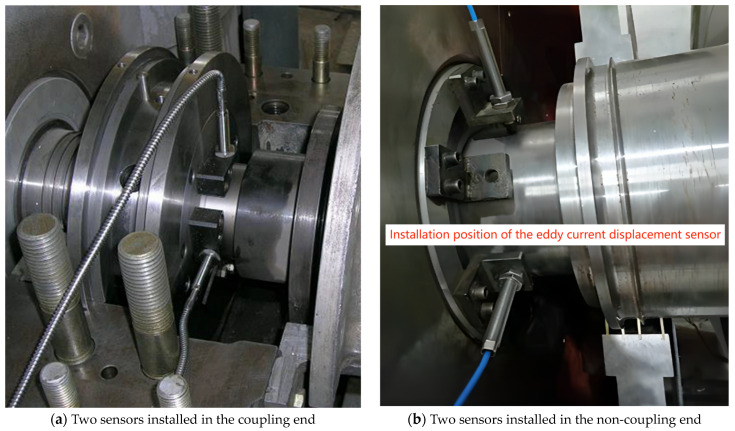
Eddy current displacement sensors positioned on the rotor, with (**a**) indicating sensors at the coupling end and (**b**) at the non-coupling end.

**Figure 3 sensors-24-04123-f003:**
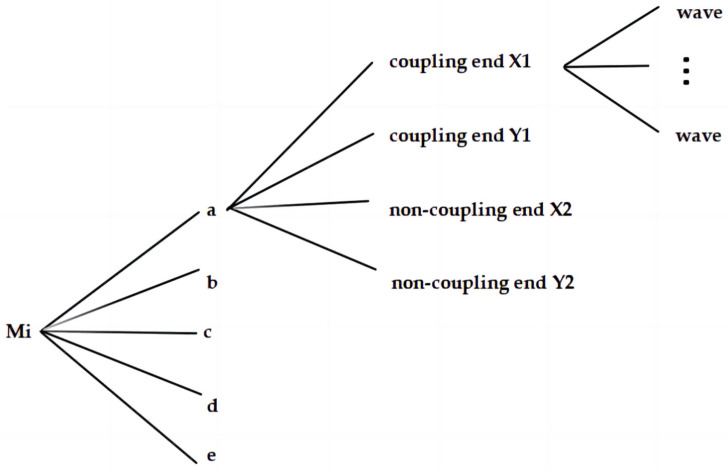
Dataset structure for rotors M1 to M16, where ‘i’ ranges from 1 to 16.

**Figure 4 sensors-24-04123-f004:**
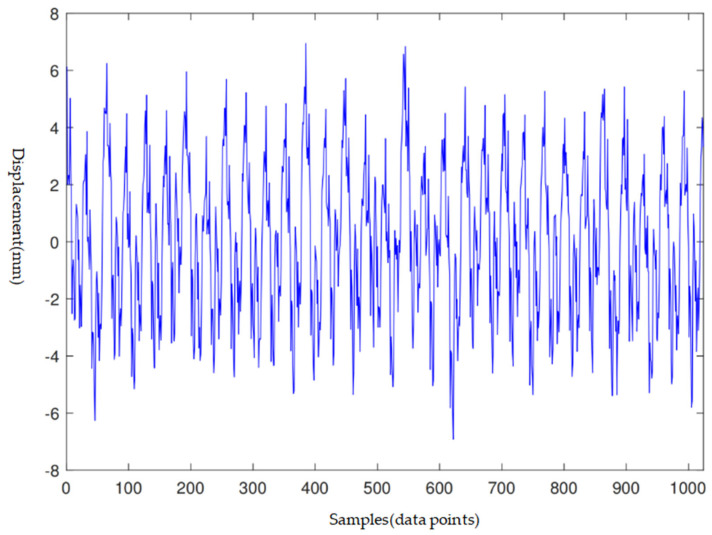
Wave data time history, exemplified by a wave from the X1 direction at the coupling end of rotor M1 during the ‘a’ time segment.

**Figure 5 sensors-24-04123-f005:**
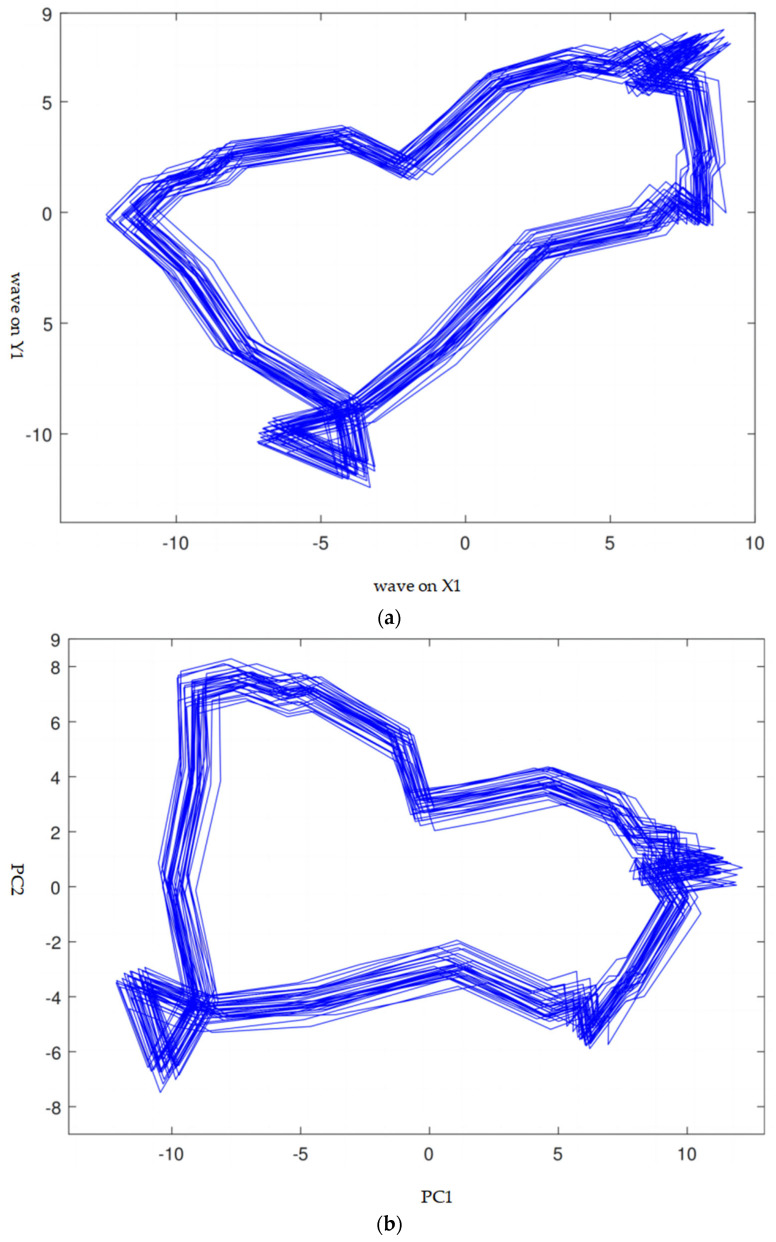
The orbits of the coupling end of M9 at the ‘a’ time segment (**a**) before and (**b**) after PCA.

**Figure 6 sensors-24-04123-f006:**
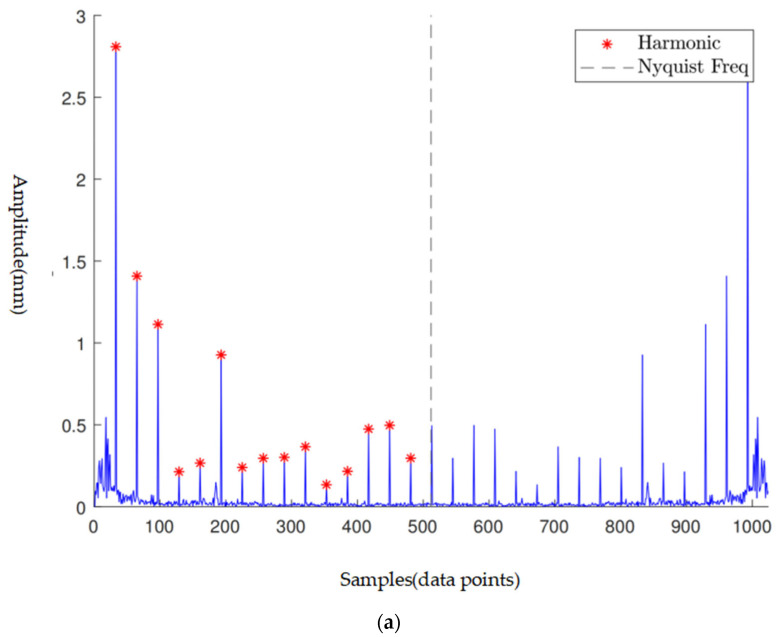
The spectrum of (**a**) the normal rotor and (**b**) the faulty rotor.

**Figure 7 sensors-24-04123-f007:**
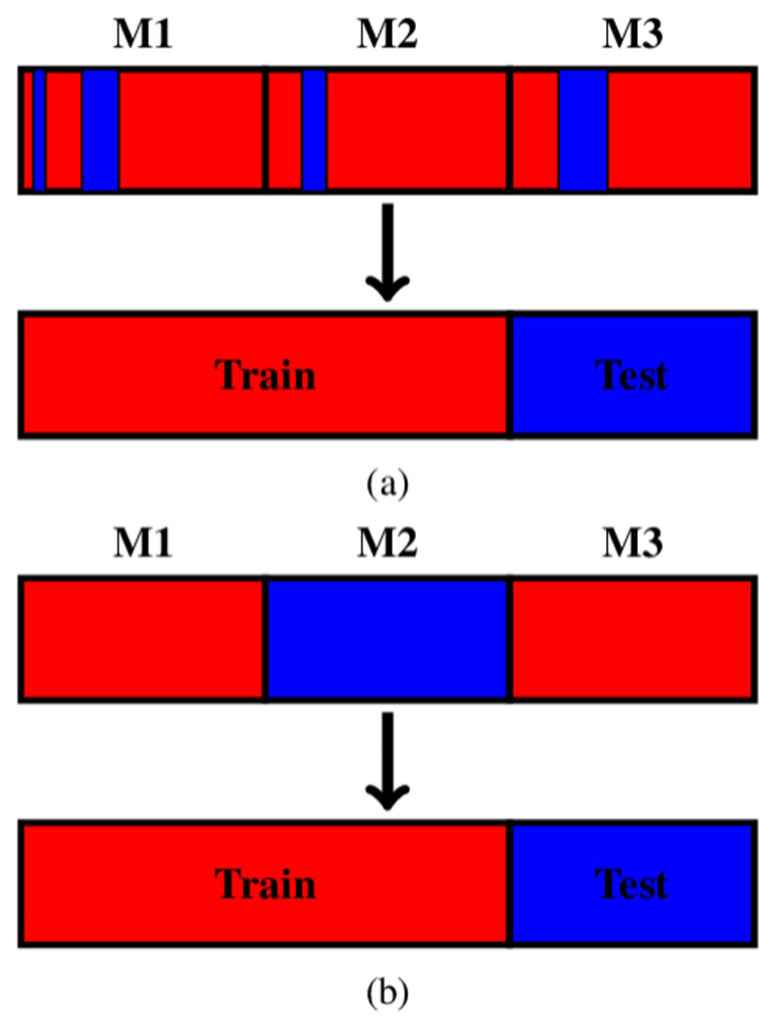
The comparison between (**a**) the traditional cross-validation and (**b**) the modified cross-validation.

**Figure 9 sensors-24-04123-f009:**
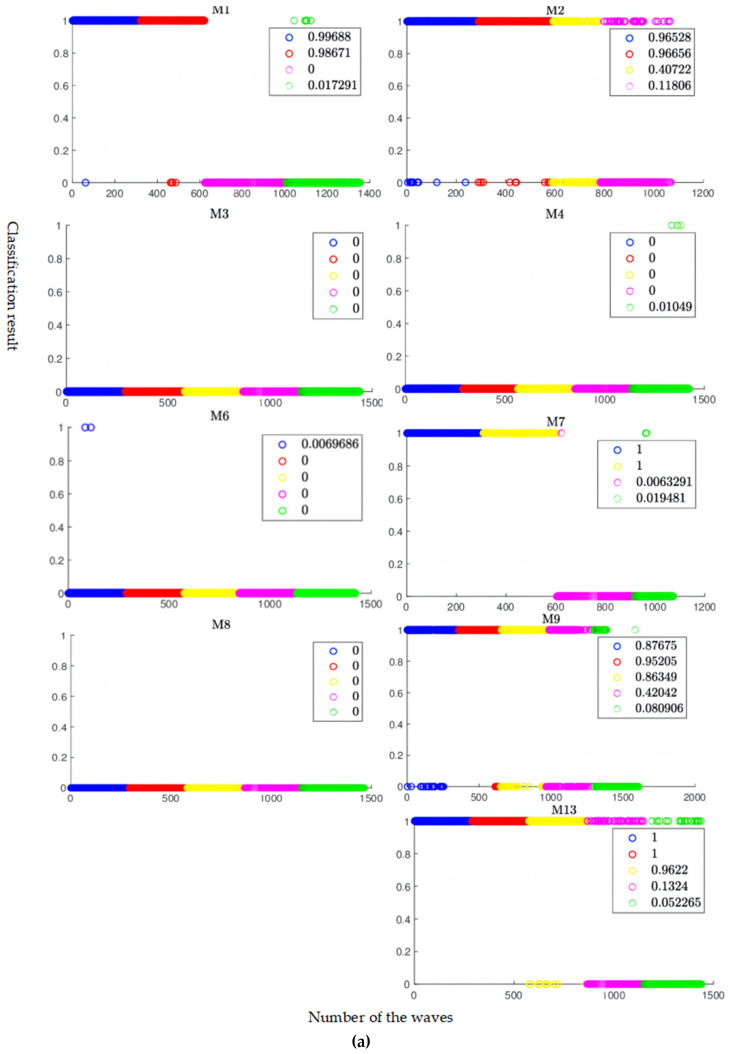
The classification results of the LDA model on (**a**) the training data and (**b**) the testing data. In time order, the blue circles are the results for the waves in ‘a’ time segment, red is for ‘b’, yellow is for ‘c’, magenta is for ‘d’, and green is for ‘e’. The classification results ‘1’ and ‘0’ represent the faulty and normal groups, respectively. The legend shows the proportion of the waves in the faulty group.

**Table 1 sensors-24-04123-t001:** Basic parameters of electric eddy current displacement sensor.

Type	NCDT3005-U3-A-C1
Full Scale Error (μm)	±2.5
Measuring Range (mm)	3
Resolution Ratio (μm)	0.1 (static state), 0.2 (dynamic state)
Frequency Response (Hz)	25K (−3 dB)
Temperature Drift (%/°C)	0.04

**Table 2 sensors-24-04123-t002:** Main performance parameters of the NI PCI-6221 data acquisition card.

Analog input channels	16
Analog output channels	2
Maximum sampling rate (kHz)	250
Voltage input range (v)	±10, ±5, ±1, ±0.2
Accuracy (μV)	100 (±10 V)

**Table 3 sensors-24-04123-t003:** The minimum and the maximum of the 13th harmonic amplitude of the non-coupling end of PC2.

	Rotor	Min	Max
Faulty	M1	12	152
	M15	62	203
Normal	M3	75	145
	M8	0	67

**Table 4 sensors-24-04123-t004:** Datasets used for feature selection and classification model training.

Rotor	Time Segment Faulty Normal	
M1	a,b,c	d,e
M2	a,b,c	d,e
M7	a,b,c	d,e
M9	a,b,c	d,e
M13	a,b,c	d,e
M3		a,b,c,d,e
M4		a,b,c,d,e
M6		a,b,c,d,e
M8		a,b,c,d,e

**Table 5 sensors-24-04123-t005:** The 10 optimal features selected from the 64 candidates by the modified cross-validation.

coupling PC1	7th and 14th harmonics
coupling PC2	2nd, 8th, and 10th harmonics
non-coupling PC1	2nd and 12th harmonics
non-coupling PC2	1st, 2nd, and 15th harmonics

**Table 6 sensors-24-04123-t006:** The status of the rotors for training and testing.

Faulty	Normal
Train M1, M2, M7, M9, M13 Test M10, M11, M15	M3, M4, M6, M8 M12, M14, M16

**Table 7 sensors-24-04123-t007:** The abnormal waves in the time segments.

Rotor	Abnormal Wave
M1	all waves in c
M2	the first 200 waves in c, all waves in e
M5	all waves in a,b,c,d,e
M7	all waves in b

## Data Availability

The data that support the findings of this study are not publicly available due to owner.

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
