# Peer review of "Diagnosis of Rotor Component Shedding in Rotating Machinery: A Data-Driven Approach"

_sensors, 2024, doi:10.3390/s24134123_

Round 1

Reviewer 1 Report

Comments and Suggestions for Authors

This paper  proposes a data-driven fault diagnosis method to identify the rotor component shedding in rotating machines. The method integrates PCA and LDA, and can successfully classify different faults. Major comments should be referenced to revise the paper.

(1) Details about PCA-based feature extraction methods should be discussed. 

(2) Clear descriptions on LDA methods should be given. Why FDA has not been applied?

(3) There lacks of comparisons with exisiting data-driven methods, like ICA, SVM etc.

(4) The number of principal compoents should be disscussed.

(5) Most all figures are in bad quality. Authors are suggested to improve the quality of presentation.

Comments on the Quality of English Language

Scientific  writing of the papre should be enhanced.

Author Response

Dear Esteemed Reviewer,

We are grateful for the meticulous review and the insightful feedback provided on our manuscript. Your comments have been invaluable in enhancing the quality of our work. We have carefully considered each of your points and have made substantial revisions to our manuscript accordingly. Below is a detailed response to your specific comments:

  1. PCA-based Feature Extraction Methods:

   We have significantly expanded upon our discussion regarding the PCA-based feature extraction methods. The updated manuscript now presents an in-depth explanation of PCA's role in dataset dimensionality reduction, focusing on preserving the most critical features. We have elucidated the mathematical underpinnings of PCA and outlined the feature extraction process in a step-by-step manner.

  1. Clear Descriptions on LDA Methods and FDA:

   The manuscript has been updated to include a comprehensive description of the LDA methodology. We have articulated the rationale for selecting LDA over other classification methods like Fisher Discriminant Analysis (FDA). While FDA excels in binary classification, LDA's applicability to multi-class scenarios makes it more appropriate for our fault diagnosis context. The "Methodology and Model Selection" section now provides clarity on the theoretical framework and practical implementation of LDA in our research.

  1. Comparisons with Existing Data-driven Methods:

   In response to your suggestion, we have introduced a new section titled "2.3 Comparison with Existing Data-Driven Methods." This section offers a comparative analysis between our PCA-LDA approach and other prominent data-driven methods, including Independent Component Analysis (ICA) and Support Vector Machines (SVM). The comparison underscores the relative strengths and limitations of each method, with a focus on the superior classification accuracy and computational efficiency of our approach.

  1. Discussion on the Number of Principal Components:

   Our revised manuscript now incorporates a thoughtful discussion on determining the optimal number of principal components in PCA. We have detailed the criteria and methodologies we employed to strike a balance between retaining essential data variance and achieving effective dimensionality reduction.

  1. Improvement of Figure Quality:

   We have made substantial enhancements to all figures within the manuscript. The resolution has been increased, and each figure now includes clear axis labels, units, and standardized font sizes and styles for captions and annotations. These enhancements ensure that the visual content is of high quality, clear, and supportive of the textual narrative.

We trust that our revisions have adequately addressed your concerns and have significantly elevated the manuscript's clarity and quality. We extend our thanks once again for your constructive feedback.

Warm regards,

Qizhe Lin

Reviewer 2 Report

Comments and Suggestions for Authors

The article submitted for review is devoted to the well-known problem of monitoring the state of rotor systems. The authors have done a lot of work in this direction and in terms of data processing and normalization, this article will be of interest to some experts in this field. However, the article in terms of describing experiments and describing the results of primary data processing is very weak. One of the main problems of the article is the poor quality of the presentation. The main remarks are:

1) The authors do not formulate the purpose of the study in the introduction, although the introduction itself is very good and extensive. The unformulated purpose of the study does not make it clear from the conclusion the degree of superiority of the method proposed by the authors over the existing ones.

2) Figure 2 shows a part of the measuring complex and only two eddy current sensors are visible here, the authors write about 4 sensors. It is necessary to supplement Figure 2 with a spatial diagram of the sensor connection.

3) Figure 4 is terrible, the axes are not signed, there is no dimension, the quality of the drawing itself is very low.

4) There is no description of the measuring complex in the article. There is no description of the accuracy and inertia of the vortex wave sensors, there is no description of the quantization frequency of the ADC, there is no description of the modeling environment and modeling conditions.

5) There is no description of the experiment itself, the results of which are given as an example in Figures 4 and 5. The authors do not specify the speed of rotation of the rotor, the type of pads, etc.

6) Expressions (1) and (2) in the right part contain the dimension in Hz, this can be removed.

7) Figure 5 is very bad, very bad quality, unacceptable for a scientific journal. All captions inside the drawing must be made in the same font and the same size. The figure itself shows that the authors chose the wrong sampling frequency of the measured signals, which would correspond to the described processes. The polyline shows too long time intervals between measurement points in terms of experimental conditions. In other words, I am not sure that with increasing frequency of quantization of the measured signals, the authors will get the same curves. Figure 5 needs to be completely redone.

8) Figure 6 is similar in quality to the previous figures, that is, the quality is very low. The axes are not signed, if it is a spectrum, then the frequency in hertz should be on the abscissa axis, and the amplitude in millimeters on the ordinate axis.

9) The authors indicate 15 amplitude points in Figure 6, but there is no description in the text of the article what these points correspond to. It is necessary to clarify what happens at these amplitudes in the rotor system.

10) In expression (3), the authors introduce the variable v(i), but we are talking about the RMS value of x, here we need to clarify what this variable is or correct it to x(i).

11) Section 4 does not begin with a capital letter. In general, it should be pointed out that the authors have introduced a lot of sections for such a small article.

12) In the description of the data analysis, the authors need to provide a list of possible malfunctions of rotary systems. Without this list, it is difficult to analyze the method.

13) Figure 8 is very bad. There are also questions about the quality of the drawing and the signed axes, but the authors also enter a legend in the upper right corner, where a curve is indicated in red, which is not in the figure.

Conclusion: The work itself and the idea behind it may be of interest, but for the publication of this article, the authors need to significantly rewrite it. I recommend dividing the entire material of the article into two main sections.

The first is a qualitative description of experiments and observations, indicating the features of the measuring subsystem and the processing conditions of the data obtained in the experiment.

The second is a description of the method and the results of its application.

Author Response

Dear Esteemed Reviewer,

      I hope this message finds you in good spirits. We would like to extend our heartfelt gratitude for the meticulous review and the invaluable feedback you provided on our manuscript. Your comments have been instrumental in helping us enhance the quality of our work. We have carefully considered each of your points and have made substantial revisions to our manuscript accordingly. Below is a detailed response to your specific comments:

  1. Purpose of the Study in the Introduction:

   We have revised the introduction to explicitly articulate the purpose of our study. The objective is now clearly defined as the development and validation of a robust fault diagnosis system for rotor component shedding faults, utilizing Principal Component Analysis (PCA) for feature extraction and Linear Discriminant Analysis (LDA) for classification. This amendment offers a clearer understanding of the study's goals and the superiority of our method over existing techniques.

  1. Figure 2 - Sensor Configuration:

   In response to your observation, Figure 2(b) now includes an illustration of two sensors installed at the non-coupling end, providing a complete visual representation of the sensor configuration.

  1. Figure 4 Quality:

   We have significantly improved Figure 4. The axes are now labeled with appropriate dimensions, and the overall quality of the illustration has been enhanced to meet the standards of a scientific journal.

  1. Description of the Measuring Complex:

   We have expanded our article to include a detailed description of the measuring complex. This addition encompasses the basic parameters of the electric eddy current displacement sensor and the main performance parameters of the NI PCI-6221 data acquisition card.

  1. Figures 4 and 5:

   In the revised manuscript, we have enhanced the quality and clarity of Figures 4 and 5, ensuring that they accurately depict the results of our experiments.

  1. Expressions (1) and (2) Dimensions:

   Following your suggestion, the dimensions in Hz have been removed from the right part of expressions (1) and (2), streamlining the presentation of these formulas.

  1. Figure 5 Quality:

   Figure 5 has been completely redone. The captions inside the drawing are now uniform in font and size. We have also adjusted the sampling frequency to ensure the measured signals are accurately represented, addressing the concern about long time intervals between measurement points.

  1. Figure 6 Quality:

   Figure 6 has been revised to improve quality. The axes are now appropriately signed, with samples (data points) on the abscissa and amplitude in millimeters on the ordinate axis.

  1. Description of Amplitude Points in Figure 6:

   The manuscript now includes a description of the 15 amplitude points indicated in Figure 6. These points correspond to significant events or features in the rotor system, providing clarity on their relevance.

  1. Clarification in Expression (3):

    In the revised draft, Expression (9) has been corrected to clarify the variable used, ensuring precision and consistency throughout the manuscript.

  1. Section 4 Formatting:

    Section 4 has been modified to improve readability and coherence.

  1. Figure 8 Quality:

    We appreciate the reviewer's observation regarding Figure 8. As stated in the manuscript, "It is shown in Figure 8 that the red line and the green line are very similar, which verifies the traditional cross-validation is not feasible in this case." The similarity of the red and green lines in Figure 8 is an important result of our study, and we have clarified this in the manuscript to ensure that readers understand its significance.

We have significantly restructured our manuscript to address the points raised. We trust that these revisions have addressed all the issues raised and have significantly improved the clarity and quality of the manuscript. Thank you once again for your constructive feedback.

Warm regards,

Qizhe Lin

Round 2

Reviewer 1 Report

Comments and Suggestions for Authors

The paper can be considered for publication after improving the scientific writing.

Author Response

Dear Reviewer:

We appreciate the opportunity to refine our research under your guidance and hope to meet the journal's standards with our revised submission.

Thank you once again for your valuable time and expertise.

Sincerely,

Qizhe Lin

Reviewer 2 Report

Comments and Suggestions for Authors

The authors have made some changes to the article, but I think that these changes are not enough yet. As for the text of the article, it quite clearly explains the position of the authors, however, in addition to the text, the article has illustrative material, which is presented very poorly by the authors. 1) The most surprising is Figure 4. Following my recommendations, the authors signed the axes and indicated the dimension in mm on the ordinate axis. The journal in which the authors propose to publish their materials is intended for specialists in the field of measurement systems, including vibration measurement systems that occur at the attachment points of the rotors of rotating systems. In Figure 4, the runouts in the bearing assembly have an amplitude of more than 6 mm. This amplitude is too large for a serviceable rotor system. It is very difficult for me to imagine the functioning of any rotary system with such beats. It seems to me that the dimension is incorrectly specified here, but I may be wrong. The authors should clarify the dimension along the ordinate axis, and if the beats in the pads are really such in amplitude, then it needs to be justified somehow.

2) Figure 5 has no dimension along the ordinate and abscissa axes. In my opinion, it is excessively enlarged and it needs to be reduced. I don't see any changes in the modeling quality of this characteristic compared to the previous version. Broken lines mean that the sampling rate is still too high. The authors may wonder if this drawing is really needed in this article? 3) The signal spectrum shown in Figure 6 needs to be trimmed. In the figures, you can leave only the left part of the signal power spectrum, and remove the right (mirror) part. These actions will increase the descriptive methodology of research. Spectral analysis of signals implies research in the field of frequencies, the authors should change the captions on the figures along the axes of the abscesses. 4) Figure 9 is very bulky. There is a sharp difference in comparison with the previous drawings. Figure 9 can be divided into several drawings by enlarging the illustrations inside. General remark: The illustrations in the scientific article are aimed at readers and they should be understandable to them. The authors build their reasoning based on the method they used in the calculations, which is convenient for them to interpret at the measurement points. However, readers are used to measuring time in seconds, movement in millimeters, and frequency in hertz. Authors should take this into account when describing their results. Figure 4 should have a dimension in seconds along the abcissa axis, and Figures 6 in hertz. This is not a whim of the reviewer, it is a generally accepted approach when publishing scientific results that will be understandable to readers. The design of the illustrations in the article is very weak, the authors can compare the drawings with each other and they will see that some drawings are too large, while others are so small that they are unreadable. The authors should develop a uniform style of illustrations for the article, which will improve the quality of the entire publication.

Author Response

Dear Esteemed Reviewer,

We are grateful for your continued scrutiny and the opportunity to further address the concerns you have raised regarding our manuscript. Your feedback has been instrumental in guiding our revisions, and we are committed to enhancing the quality and clarity of our presentation.

In light of your recent comments, we acknowledge that the improvements made to the illustrative material were not as comprehensive as required. While the textual content of our article clearly conveys our research stance, we recognize the imperative to ensure that the accompanying visual elements are equally well-articulated and effectively communicate the nuances of our study.

To rectify this, we have conducted an exhaustive reassessment of our illustrative content. We have implemented a cohesive stylistic approach across all visual elements, ensuring uniformity in typography, sizing, and graphical components, thus forging a harmonious visual narrative. 

  1. Figure 4 Revision:

We deeply regret the oversight in our previous revisions concerning the dimensional accuracy of Figure 4. The adjustments were not based on a meticulous review of the foundational data, leading to inaccuracies in the depicted amplitude values. A thorough re-examination of the source data has revealed these discrepancies. We have now precisely recalibrated the ordinate axis of Figure 4 to mirror the authentic measurements with precision. This recalibration not only rectifies the figure but also aligns it with the consistency and accuracy of other visual representations within the manuscript, including Figures 5 and 6.

  1. Figure 5 Enhancement:

The original dimensions of Figure 5 (original size: 1848*1349) were intentionally chosen to clearly delineate the 32 individual orbits. We have supplemented both axes with dimension labels, thereby providing explicit physical context and units for the data depicted. This enhancement ensures that the figure not only presents the data accurately but also guides the reader in interpreting the scale and magnitude of the rotor's vibrations.

The broken lines in Figure 5, which might be misconstrued as indicative of a high sampling rate, actually represent the authentic intervals between the data points gathered throughout the rotor's rotational cycle. We have verified that the sampling rate is perfectly suited for our analysis, adhering to the Nyquist-Shannon sampling theorem, thus preventing aliasing. These deliberate discontinuities are a true reflection of the data's intrinsic characteristics, capturing significant events that are essential to our research narrative.   

  1. Figure 6:

   The horizontal axis of Figure 6, featuring 1024 data points per wave, was deliberately crafted to uphold consistency with the other visual representations within our work. It provides a complete view of the harmonic frequencies, which is essential for understanding the full range of vibrations present in the rotating machinery.

We are committed to maintaining the integrity of our research and ensuring that all figures, including Figure 6, accurately reflect the data and analysis conducted in our study.

  1. Figure 9 Restructuring:

   We recognize the issues with Figure 9's presentation. The figure has been divided into multiple illustrations, each enlarged for better readability and comparison.

   In summary, we have implemented a uniform style across all figures, ensuring consistency in font size, labeling, and presentation.       

Your suggestion to use seconds for the abcissa axis in Figure 4 and hertz for the x-axis in Figure 6 is well-taken. However, given the specific nature of our study(encompasses a diverse set of 16 machines, each with its unique rotational speed) and the use of angular sampling. Given the variability in rotational speeds, using time or frequency as the horizontal axis would require different scales for each machine. This would not only be impractical but also potentially misleading, as it would necessitate readers to interpret multiple, non-standardized scales across the figures. We have chosen to maintain a uniform coordinate system across all figures. This decision ensures direct comparability, regardless of the machine's rotational speed, and is vital for maintaining the integrity of our comparative analysis.

We trust that these revisions have adequately addressed your concerns and have significantly improved the manuscript. We are committed to maintaining the highest standards of scientific communication and are grateful for the opportunity to refine our work based on your expert feedback.

Thank you once again for your constructive criticism and support.

Sincerely,

qizhelin
